# The movements of a recently urbanized wading bird reveal changes in season timing and length related to resource use

Anjelika Kidd-Weaver[1¤a], Jeffrey Hepinstall-Cymerman[1¤b]*, Catharine N. Welch[1¤b], Maureen H. Murray[1¤c], Henry C. Adams[1,2], Taylor J. Ellison[1,2], Michael J. Yabsley[1,2], Sonia M. Hernandez[1,2]

1 Warnell School of Forestry and Natural Resources, University of Georgia, Athens, Georgia, United States of America, 2 Southeastern Cooperative Wildlife Disease Study, College of Veterinary Medicine, University of Georgia, Athens, Georgia, United States of America

¤a Current address: Department of Forestry and Environmental Conservation, Clemson University, Clemson, South Carolina, United States of America
¤b Current address: Florida Department of Environmental Protection, Division of Recreation and Parks, Okeechobee, Florida, United States of America
¤c Current address: Davee Center for Epidemiology and Endocrinology, Urban Wildlife Institute, Chicago, Illinois, United States of America
* jhepinstall@warnell.uga.edu

**Data Availability Statement:** The data is publicly available from Movebank. The reference is: Kidd-Weaver A, Hepinstall-Cymerman J, Welch CC,

## Abstract

The American White Ibis (*Eudocimus albus*) is a nomadic wading bird that is increasing the amount of time spent foraging in urban areas, relying on artificial wetlands and other anthropogenic resources year-round. In this study, we explore whether and how American White Ibis association with urban environments is predictive of variation in the timing and length of behavioral seasons. Other urbanized species exhibit altered annual cycles such as loss of migratory behavior and year-round breeding related to consistent resource abundance, often related to intentional and unintentional provisioning. To determine if these same patterns of behavior were also present in White Ibis, we used behavioral change point analysis to segment the tracks of 41 ibis equipped with GPS backpacks to identify the initiation and duration of four behavioral seasons (non-breeding, pre-breeding, breeding, post-breeding) the degree of urban association. We found that intraspecific variation in urban habitat use had strong carryover effects on the timing and duration of behavioral seasons. This study revealed ibis with higher use of urban habitats in non-breeding seasons had longer non-breeding seasons and shorter breeding seasons that began earlier in the year compared to ibis that primarily use wetland habitats. The timing and duration of seasons also varied with ibis age, such that ibis spent more time engaged in breeding-related seasons as they aged. Juvenile and subadult ibis, though considered to be reproductively immature, also exhibit behavioral shifts in relation to breeding seasons. The behavioral patterns found in this study provide evidence that ibis are adapting their annual cycles and seasonal behaviors to exploit urban resources. Future research is needed to identify the effect of interactions between ibis urban association and age on behavioral season expression.

Murray MH, Adams HC, Ellison TJ, Yabsley MJ, Hernandez SM (2020) Data from: The movements of a recently urbanized wading bird reveal changes in season timing and length related to resource use. Movebank Data Repository. https://doi.org/10.5441/001/1.8ms50757.

**Funding:** Funding was provided by a National Science Foundation EEID grant, (DEB-1518611) and the Georgia Ornithological Society. The funders had no role in study design, data collection and analysis, decision to publish, or preparation of the manuscript

**Competing interests:** The authors have declared that no competing interests exist.

## Introduction

Annual cycles and movements of animals are often related to abiotic environmental conditions such as day length and temperature, temporal and spatial patterns of resources, as well as the behavioral state of the individual [1, 2]. Human activities modify local abiotic conditions and resource availability, and since conditions experienced during one portion of the annual cycle may carry over into subsequent seasons, full annual cycle studies are needed to understand the fitness effects of behavioral changes relating to human modification of the landscape. Animals can exhibit either migratory and/or nomadic behaviors in response to variable resource availability; conversely, they can also become residents in a given area by changing their resource use, rather than location [3–7]. A more accurate measure of season initiation, or the change in seasonal state (e.g., non-breeding to breeding) is required to understand how populations are modifying their movements both temporally and spatially in response to altered land use.

Without direct knowledge of an animal's behavior, seasons for a given species are typically defined based on calendar date or relationships with biotic (e.g., vegetation phenology) or abiotic (e.g., wet-dry cycles) factors derived from observational data. However, spatial, temporal, and individual variation can all reduce the accuracy of seasons derived from climatic factors and historical records alone [2, 3, 8, 9]. For example, many environmental cues, such as spring greenup and wetland water depths, vary across latitudinal gradients and also temporally; this can result in some populations of the same species transitioning between different behavioral states at different times [6, 10]. Additionally, due to fluctuations in resource availability amongst land-use types (i.e., human-altered versus unaltered), individuals living in urban areas can vary in their seasonality and exhibit different behaviors from those living in unaltered areas such as: reduced site fidelity, reduced range size, abandoned migratory movements, or year-round breeding [3, 5, 9, 11–14].

Direct observations of daily movements and multi-day patterns can be used to infer "behavioral seasons" where the start and end of a season is derived locally with consideration for differences in experiences (e.g., age, local land use/land cover composition), rather than applied across the entire range of a given species. The behavioral state of individuals from a local population is inferred from their movement patterns rather than an arbitrary calendar date or highly variable biotic signals. For example, an animal may exhibit high levels of tortuosity, the tendency to move in directions perpendicular to the current movement path, while attending a nest or while using the same resource repeatedly, or they might exhibit linear movements between disparate ranges or exhibit exploratory movements that take on a nomadic pattern before and after breeding and young rearing. Determining behavioral seasons, based on animal behavior, is critically needed to advance our understanding of movement ecology of animals [6, 8].

In this study, we explore whether and how association with urban environments is predictive of changes in behavioral seasons. We explore this idea using a highly mobile nomadic wading bird, the American White Ibis (*Eudocimus albus*) in southern Florida. Ibis, like many other species, are responding to human habitat modification with changes in movement and resource selection. Ibis in this region move to follow ephemerally available food resources and generally exhibit four identifiable types of movements associated with different seasons (details under *Study System* below). The objective of our study was to determine if and how white ibis in this area change their annual seasonal expression based on experience with wetland and urban land uses. Specifically, we predicted that the timing of seasons (season initiation) and length of each season (season duration) would differ with the degree of urban habitat association. To identify behavioral seasons for each ibis, we performed behavioral change point analysis [15, 16] on GPS locations from backpack transmitters obtained for 41 ibis over multiple

years. To examine urban habitat association, we used daytime GPS locations during the least constrained season, the non-breeding season, to classify our birds into low, medium, or high use of urban habitats. Our findings have implications for other mobile species that are altering movements and behaviors in response to land use change and increasing urbanization worldwide.

# Materials and methods

## Study system

The American White Ibis (*Eudocimus albus*) is a medium-sized, nomadic, freshwater and estuarine wading bird; however, its presence in urban areas in southern Florida, U.S.A, is thought to be increasing. Ibis are considered a nomadic species, and as such, they do not exhibit true migrations between predictable ranges. The timing of their annual cycles and movements varies related to fluctuations in hydrologic regimes: specifically, shallow water with dense prey populations for optimal foraging opportunities [17, 18]. Generally, the annual cycle consists of four seasons: active breeding from egg-laying to young independence; pre- and post-breeding migrations in which ibis move directly or nomadically between distinct geographic areas; and non-breeding seasons in which ibis move nomadically between foraging areas without restrictions related to reproduction. During the non-breeding season, ibis movement behaviors are most flexible, and they frequently change foraging and roosting sites as they follow variable water depths in both fresh and saltwater wetlands. In contrast, during reproduction, adults restrict foraging to freshwater wetlands in proximity to mixed-species rookeries formed on treed islands to support the growth of nestlings and energetic needs of adults [18–20]. Freshwater wetlands are selected because young ibis cannot excrete salt as well as adults [19] and treed islands capitalize on protection afforded by American alligators (*Alligator mississippiensis*) against terrestrial predators [21].

In recent decades, ibis living in southern Florida have increased their use of urban habitats, due to massive wetland loss from agricultural and urban land conversion and degradation of remaining wetland habitats. While living in urban areas, ibis can exploit anthropogenic resources and exhibit resident behaviors as the need to search for foraging opportunities diminishes [22–24]. These changes in behaviors may lead to less knowledge of environmental conditions in areas outside of urban habitat and a concomitant lengthy search for suitable breeding sites.

## Study site: Palm Beach County, Florida

Palm Beach County, Florida provides an opportune location to study the effects of urbanization on the American White Ibis because of its relative rapid urbanization and growth. Palm Beach County is Florida's third most populous county, with approximately 1.4 million people. Within this county, 55% of the human population lives in only 16.6% of the county's land area, which primarily consists of coastal incorporated urban areas. The remaining 83.4% of Palm Beach County's area is unincorporated land, primarily composed of residential areas (15.4%), agricultural lands (40.6%) and natural lands (44%) [25]. Urbanized, densely populated areas are juxtaposed against natural wetland areas and agricultural lands, which provide ibis with numerous foraging and roosting resource options on both a daily and seasonal basis. Within Palm Beach County, we chose 15 capture sites that represented a range of urbanization from urban parks where ibis are actively provisioned by human visitors, to large wetland complexes where ibis have little contact with humans (Fig 1).

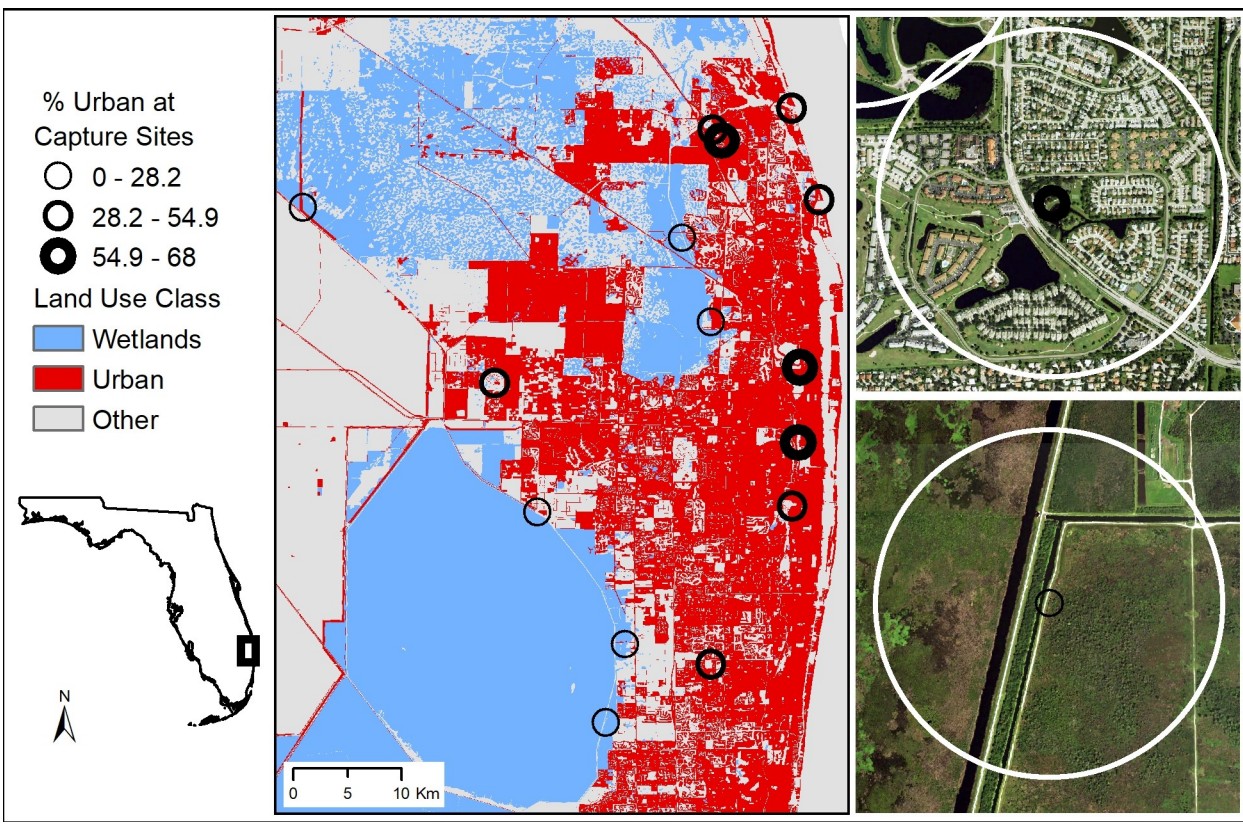

**Fig 1. Ibis capture sites in Palm Beach County, Florida.** Left panel shows reclassified land use from the Cooperative Land Cover (CLC version 3.2) map. Symbol line width indicated percent of urban land use within a 650-meter radius ranging from 68% urban residential areas and urban parks (upper right panel) to 0% urban, wetland landscapes (lower right panel) as depicted by recent aerial photographs.

### Ibis capture, transmitter deployment, and GPS tracking

To outfit ibis with GPS transmitters, they were captured utilizing two methods. Ibises in urban parks were captured using leg lassos or a manual flip-trap baited with bread [26, 27]. Ibises in wetlands were captured with mist nets and decoys, as they could not be baited or approached [28]. At least two individuals operated all methods to ensure quick and safe extraction of birds upon capture. All animal capture and handling procedures were reviewed by the University of Georgia's Institutional Animal Care and Use Committee (IACUC # A2016 11-019-Y2-A0), and conducted under both a Florida Wildlife Conservation Commission permit (LSSC-11-00119F) and a United States Fish and Wildlife Agency permit (MB779238-0). Once captured, ibises were aged by plumage as juveniles (e.g., 75% to 100% brown feathers), subadults (e.g., some to <25% brown feathers), or adults (e.g., no discernable brown feathers), weighed, and fitted with Ecotone GPS-GSM (2G) transmitters (North Star Science and Technology, Oakton, VA, USA) using a backpack harness [19, 29, 30]. Ibis sex was determined using polymerase-chain reaction (PCR) from blood samples taken at capture using standard methods [31]. Transmitters were only applied to ibis for which the transmitter, harness, and identification band were less than 3% of the bird's mass [32]. GPS units received up to 12 locations per day at a maximum of 2-hour intervals, primarily during daylight hours. GPS units were allocated among capture sites such that there were 2–4 deployed units per capture site.

Ibis were captured and fitted with transmitters during the following periods: October—November 2015 (n = 15), February—March 2016 (n = 17), June—July 2016 (n = 5), October—

November 2016 (n = 4), and February—March 2017 (n = 7). GPS transmissions were monitored until 8 June 2018, or until transmitter failure. For juveniles and subadults, as ibis aged through the duration of their transmitter deployment, we adjusted their estimated age each spring in the deployment history until they reached adult status.

Forty-eight GPS transmitter deployments were made between October 2015 and February 2017. Ibis captures were equally distributed across an urbanization gradient from areas with 0 to 68% urban land cover within a 650-meter radius around the capture site (Fig 1). The mass of individuals ranged from 800 to 1240 grams (mean: 962.9, mean transmitter:body weight 2.67% [2.05–3.29%]). The mass requirement led to a skewed sex ratio (36 males, 12 females), and older birds (80% subadult or older at capture). Deployment from release date until transmitter failure, individual death, or program termination date (8 June 2018) ranged from 9.75 days to 948.25 days (median: 316.67 days). Of the transmitter failures, 13 occurred between 30 December 2016 and 2 January 2017 corresponding to the deactivation of ATT 2G GSM cellular networks. We removed seven individuals from the analysis due to limited available data (< 2 seasons or < 30 days), leading to an ultimate sample size of 41 individual birds.

## Urban habitat use

Daytime locations for all individuals during non-breeding seasons (as defined in behavioral change point analysis below) were used to represent the level of urban association [11] as this season and time of day represent the least constrained habitat use (i.e., can use freshwater wetlands, brackish wetlands, or urban habitats), as opposed to night time or breeding locations in which an individual's choices are constrained by specific resource needs requiring use of more natural habitats (i.e., tree-island roosts, or access to freshwater only foraging). Non-breeding urban habitat use was summarized using the 2016 Cooperative Land Cover (CLC version 3.2; https://www.fnai.org/LandCover.cfm) map for the state of Florida, a 10-meter resolution raster geospatial layer with 234 land cover classes. For this study, we were interested in differentiating between urban and wetland habitat use, so we reclassified land cover classes into urban, freshwater wetland, and other. We defined urban habitat use as the mean proportion of urban land cover within a 650-meter radius of daytime non-breeding locations. The 650-meter radius was derived using a first passage time analysis (FPT) to estimate the scale of ibis foraging [33]. FPT calculates the time it takes for an individual to leave a circle of fixed radius, representing the scale of different types of movements [34]. We performed FPT for all individuals to find the minimum optimal radius, which we considered the scale of an individual's localized movements, and used the median radius value (650-meters) to represent the minimum scale of habitat selection for all ibis [33] and to account for uncertainty in ibis locations within the 2-hour window of locations, use of edge habitats, or GPS error.

## Behavioral season timelines

Behavioral change point analysis (BCPA) can be used to detect changes in movement characteristics from tracking data that are difficult to interpret visually or with data structures not suitable for other techniques [15, 16]. BCPA uses moving window and likelihood methods to examine time series movement data and identify points where the underlying structure of the movement track changes, corresponding to changes in an individual's behavior. BCPA has been used to define both large and small-scale movements such as separating segments of animal migrations or identifying foraging versus resting bouts within a single day [35–39].

To classify an individual ibis's movement track into behavioral seasons, we performed BCPA separately for each ibis using two movement metrics (persistence velocity [the tendency of movement to continue in a certain direction] and tortuosity [the tendency for movement to

occur perpendicular to the current movement direction]) as recommended by Gurarie, Andrews and Laidre [15] in Program R version 3.4.3 [40] using the *bcpa* package [15, 41, 42]. We used a sub-sampling window size of 120 sequential location observations to reflect a 10-day period in which ibis will locate, exploit, and abandon a new wetland foraging site observed in a previous study [43]. The track segments between change points identified by BCPA are referred to as "bouts" and correspond to segments of the track where the parameter estimates for the movement statistic are stable, indicating the individual's movement behavior is consistent (S1 Fig). Minor change points are filtered from the BCPA by selecting significant change points from minor change points within a temporal window (10 days) using the "flat" summary in the *bcpa*.

We then categorized ibis bouts into behavioral seasons using a visual analysis of bouts in Arc-GIS 10.6.1 [44] based on time of year relative to published annual cycles [18], the pattern of locations on the landscape (e.g., linear, large cluster, several small clusters), and associations with known or potential breeding resources. Temporally adjacent bouts that were considered to be representative of the same seasonal behavior were merged. Specific decision rules followed are provided in supplemental materials (S1 File: Defining ibis BCPA bouts as behavioral seasons). Thus, each bout was assigned a behavioral season (non-breeding, search-and-dispersal, breeding, post-breeding) by considering a combination of time of year, movement characteristics (widespread versus local versus linear), and resource association (e.g., a known rookery location).

We classified each individual ibis timeline into seasons and performed analyses based on two metrics: the duration of each season; and the initiation dates for each season. The number of seasons recorded is not equal to the number of individual birds tracked because ibis had variable timelines (e.g., some birds were tracked through multiple non-breeding seasons while others were only tracked through one). We explored differences in duration and initiation date by sex, age, and between years. For each season and grouping variable of interest, we performed difference in means tests, Kruskall-Wallis for comparisons with more than two factor levels, or Two-Sample Wilcoxon Test. For grouping variables with more than two factor levels and results with p < 0.1, we used multiple pairwise comparisons tests to determine which pairs were statistically significantly different using multiple comparison Wilcoxon Tests and Benjamini and Hochberge [45] correction.

### Behavioral season differences related to non-breeding season use of urban habitats

To address our goal of exploring how individual association with urban environments may predict changes in the timing and duration of behavioral seasons, we grouped ibis by their daytime use of urban land cover/land use classes (hereafter "urban habitats") in the non-breeding season into quantiles. We then compared the amount of time (duration) individuals spent in each defined behavioral season and the initiation date of each season to identify relationships relative to the use of urban habitat of a given individual during the non-breeding season. Use of urban habitats in the non-breeding season ranged from 0.3% to 68.6% and were subsequently grouped into three classes by quantile: 0–28.2%, 28.2–54.9% and >54.9%, representing "low", "intermediate", and "high" urban-use, respectively. We tested for differences in season duration and initiation as detailed above. Comparisons were made for all seasons except for initiation date of the non-breeding season because its start timing was highly dependent on the timing of other seasons.

## Results

### Behavioral season timelines

Ibis spent on average 136 days (sd = 67, n = 62) in non-breeding seasons, 66.2 days (sd = 39.5, n = 54) in search-and-dispersal seasons, 72.5 days (sd = 46.2, n = 48) in breeding seasons, and

37.1 days (sd = 46.2, n = 40) in post-breeding seasons (Fig 2). High standard deviations relative to mean duration for search-and-dispersal and post-breeding seasons were due to several birds skipping these seasons (see below). We expected each ibis to exhibit all behavioral seasons sequentially: non-breeding, pre-breeding search-and-dispersal, breeding, post-breeding movement, and return to non-breeding. However, some seasons were not observed and these "skipped seasons" were identified where a season was missing in the sequence of an individual's movement track and have a duration of zero.

While all individuals exhibited some form of a non-breeding season when expected, we identified six skipped search-and-dispersal seasons, three skipped breeding attempts, and 14 skipped post-breeding movements across all individuals and years (Fig 2). Differences in season duration by year were only significant in the non-breeding season with 2015 shorter than 2017 (Kruskal-Wallis p = 0.05). Season initiation dates were significantly different for both search-and-dispersal and breeding seasons between years. Search-and-dispersal seasons started significantly later in 2016 than in 2017 and 2018 (Kruskal-Wallis p < 0.01), while 2017 and 2018 seasons were similar. Breeding seasons started significantly earlier each subsequent year of our study (Kruskal-Wallis p < 0.01).

Younger birds exhibited movement patterns that deviated from their typical non-breeding movement patterns, resembling breeding related behaviors of adults and identified as search-and-dispersal or breeding behavioral seasons, though with different duration and initiation (S2 and S3 Figs). For example, younger birds (< 3) had significantly longer search-and-dispersal seasons compared to both 3 and 4-year old birds. Three and 4-year old birds who exhibited search-and-dispersal seasons, started this behavior later than adults. We observed a trend that older birds had longer non-breeding seasons, 3 and 4 year old spent less time in search-and-dispersal, and 4 and 5 year olds spent longer in the breeding season.

Ibis season duration only differed by sex in the non-breeding season, with females remaining significantly longer in this season than males (S4 Fig). There were no significant differences in season initiation date between male and female ibis.

## Behavioral season differences related to non-breeding season use of urban habitats

Mean season duration was significantly different amongst urban classes in the non-breeding (p < 0.05) and suggestive of differences in search-and-dispersal (p < 0.1) seasons (Fig 3). Non-breeding seasons were significantly shorter for low urban-use ibis compared to other urban-use classes. In contrast, high urban-use ibis spent significantly less time in search-and-dispersal seasons than low urban-use ibis.

Ibis with greater non-breeding daytime use of urban habitats began their breeding season earlier than ibis with lower non-breeding daytime use of urban habitats (Fig 4). Season initiation date did not significantly differ across years for low urban-use ibis, though the effect of year was significant for those with intermediate and high urban-use in some but not all seasons. Yearly differences in initiation dates were significantly different for intermediate urban-use ibis only in the search-and-dispersal season, which started significantly later in 2016 than either 2017 or 2018 (Fig 5). For high-urban use ibis, initiation dates differed significantly in both the search-and-dispersal and breeding seasons, both starting later in 2016 than 2017 or 2018. We found no statistically significant differences between season duration or initiation by sex or age and class of urban habitat use during the non-breeding season.

Skipped seasons, for example where an animal moved directly to or from a breeding location rather than exhibiting search-and-dispersal or post-breeding behaviors, were also related to the amount of use of urban habitat with high and intermediate urban-use ibis more often

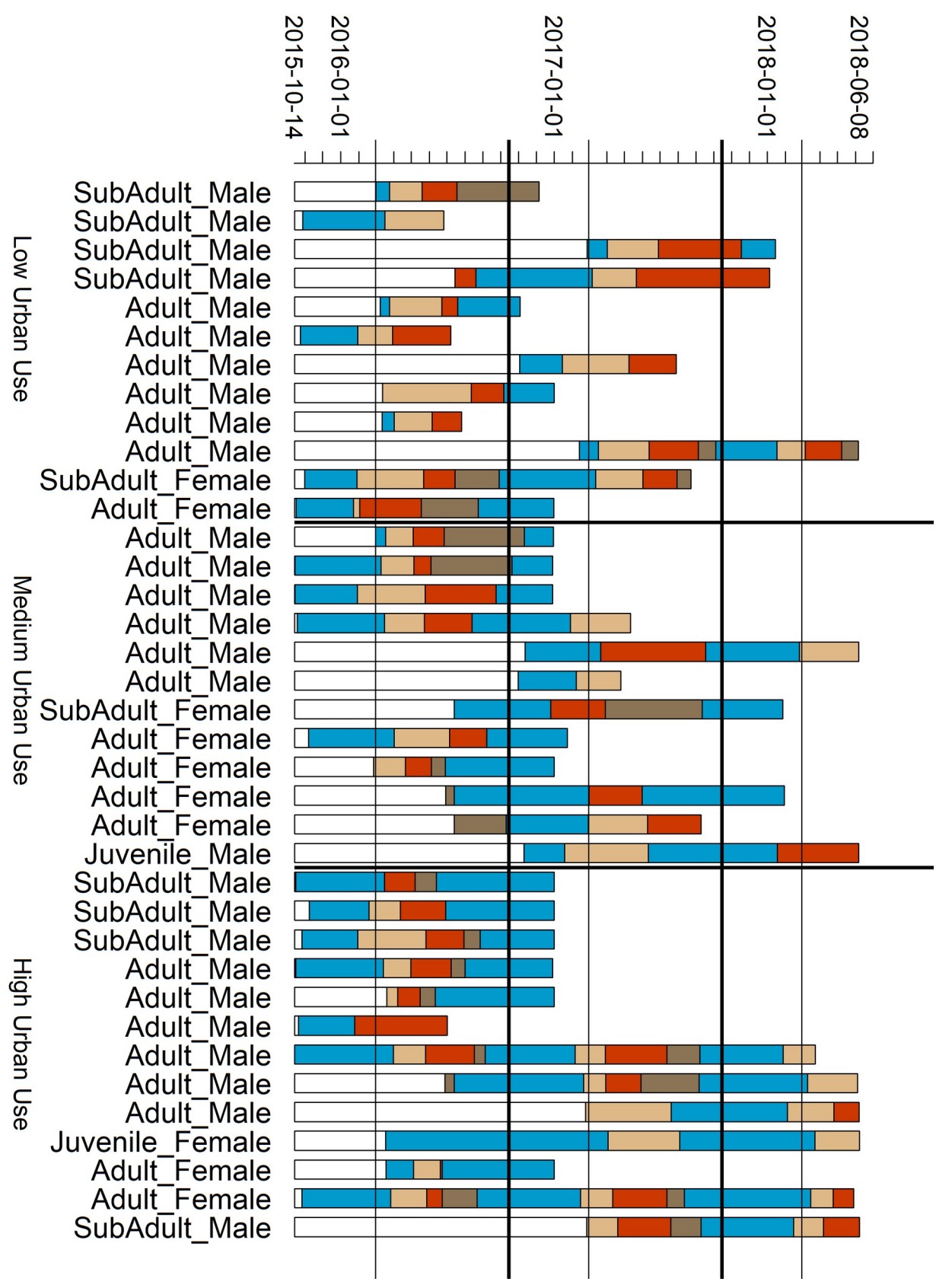

**Fig 2. Seasonal timelines for individual ibis.** Ibis are ordered from least (top) to most (bottom) use of urban habitat in the non-breeding season along the y-axis. Each coloured segment represents the temporal sequence of behavioral seasons identified for each bird: non-breeding (blue/dark grey), pre-breeding search and dispersal (light brown/loose hashed), breeding (red/light grey), and post-breeding (dark brown/dense hashed). Blank segments (white) account for delays in deployment after the first deployed transmitter. Vertical bars show the approximate beginning (thin) and end (thick) of the ibis breeding seasons from observational studies of colonies, March 1 and October 15. Horizontal lines indicate the 3-quantile cutoffs to define low, intermediate, and high use of urban habitat. Ticks on the x-axis correspond to the first day of each month with the start and end dates of the study and January 1st of each year shown for reference. Four individuals that were not tracked long enough to cover a complete non-breeding seasons are not included in this plot.

skipping search-and-dispersal and breeding seasons. Of the 6 skipped search-and-dispersal seasons, three were skipped by intermediate urban-use ibis (19% of ibis with intermediate urban use) and three by high (17%) urban-use ibis. All three of the skipped breeding seasons

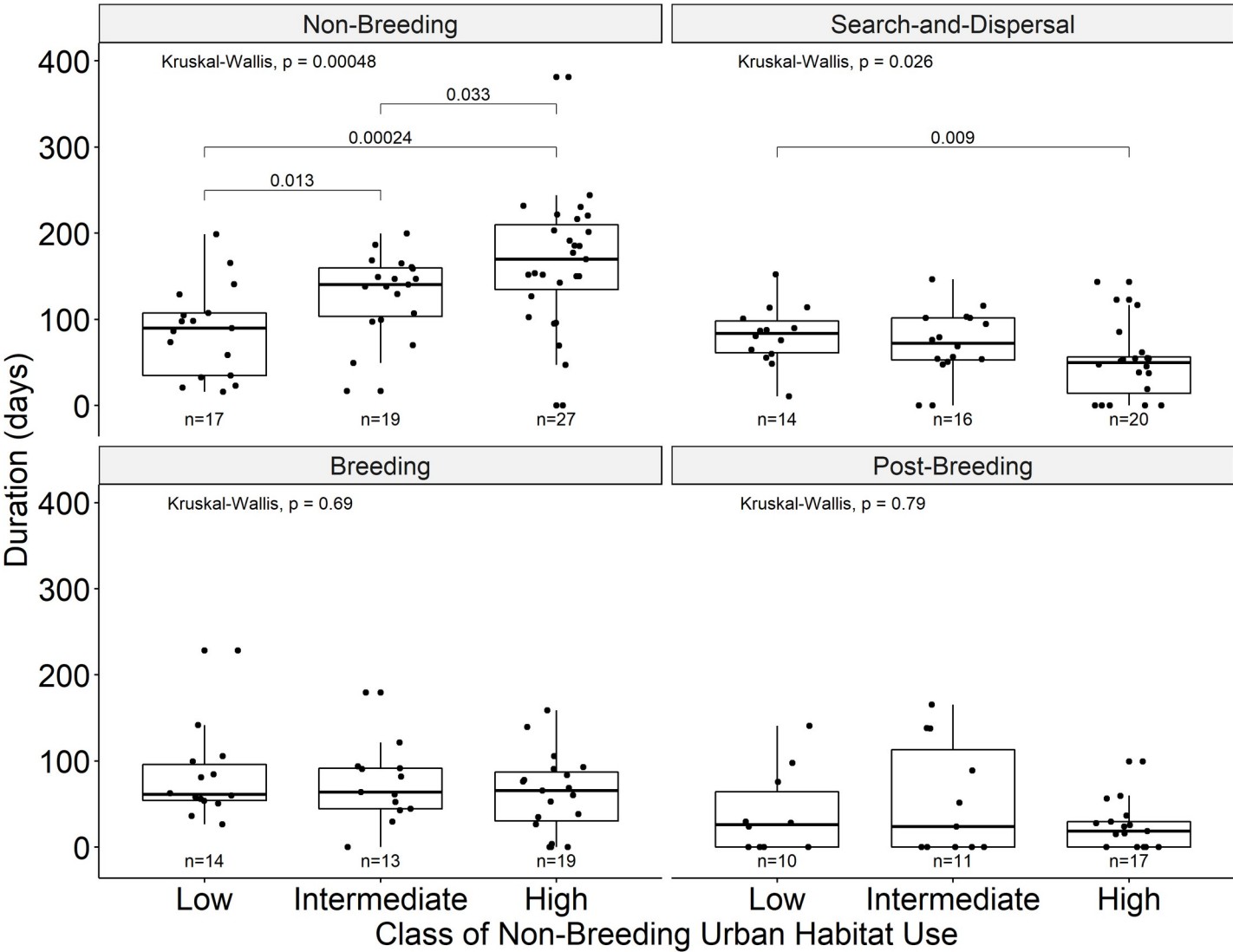

**Fig 3. Boxplots of duration of each behavioral season compared within each class (low, intermediate, high) of daytime use of urban habitat during their non-breeding season.** Four behavioral seasons (non-breeding, search-and-dispersal, breeding attempt, and post breeding) are grouped by daytime, non-breeding season urban habitat use classes. Individual values shown as points in each boxplot. Sample size for each group is indicated below plots; sample sizes are not equivalent across seasons due to differences in timing of deployment and tracking end (Fig 2). P-values for Kruskal-Wallis tests indicate difference in means within a group, with lines and p-values indicating statistically significant (p < 0.05) Wilcoxon Rank Sum test statistic for respective pairwise differences in means.

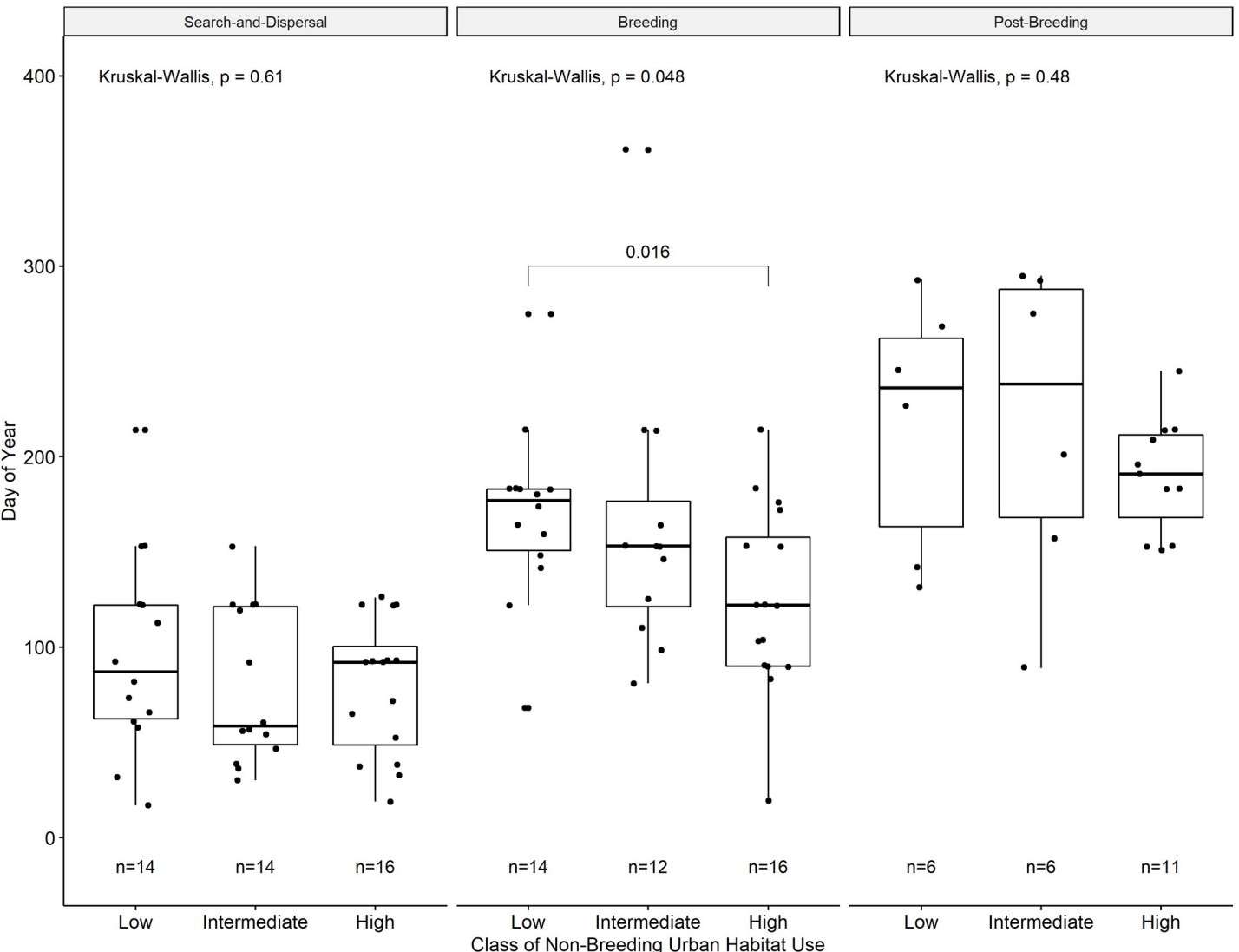

**Fig 4. Boxplots of initiation date of search-and-dispersal, breeding, and post-breeding behavioral seasons compared within each class (low, intermediate, high) of daytime use of urban habitat during their non-breeding season.** Individual values shown as points in each boxplot. P-values for Kruskal-Wallis tests indicate difference in means within a group, with lines and p-values indicating statistically significant (p < 0.05) Wilcoxon Rank Sum test statistic for respective pairwise differences in means.

were by high urban-use ibis (6%). Almost equal number of low (4; 31%), intermediate (5; 36%), and high (5; 25%) urban-use ibis skipped post-breeding seasons, instead returning directly to previously used feeding locations.

## Discussion

We found that intraspecific variation in use of urban habitat had strong carryover effects on the timing and duration of behavioral seasons in a highly mobile nomadic species. Specifically, we found that ibis vary widely in the timing and duration of seasons, which cannot be predicted simply by calendar date or previously established seasons for this species. We found that season initiation and duration are related to urban habitat use, modulated by individual traits such as age and annual variation likely related to yearly environmental variations.

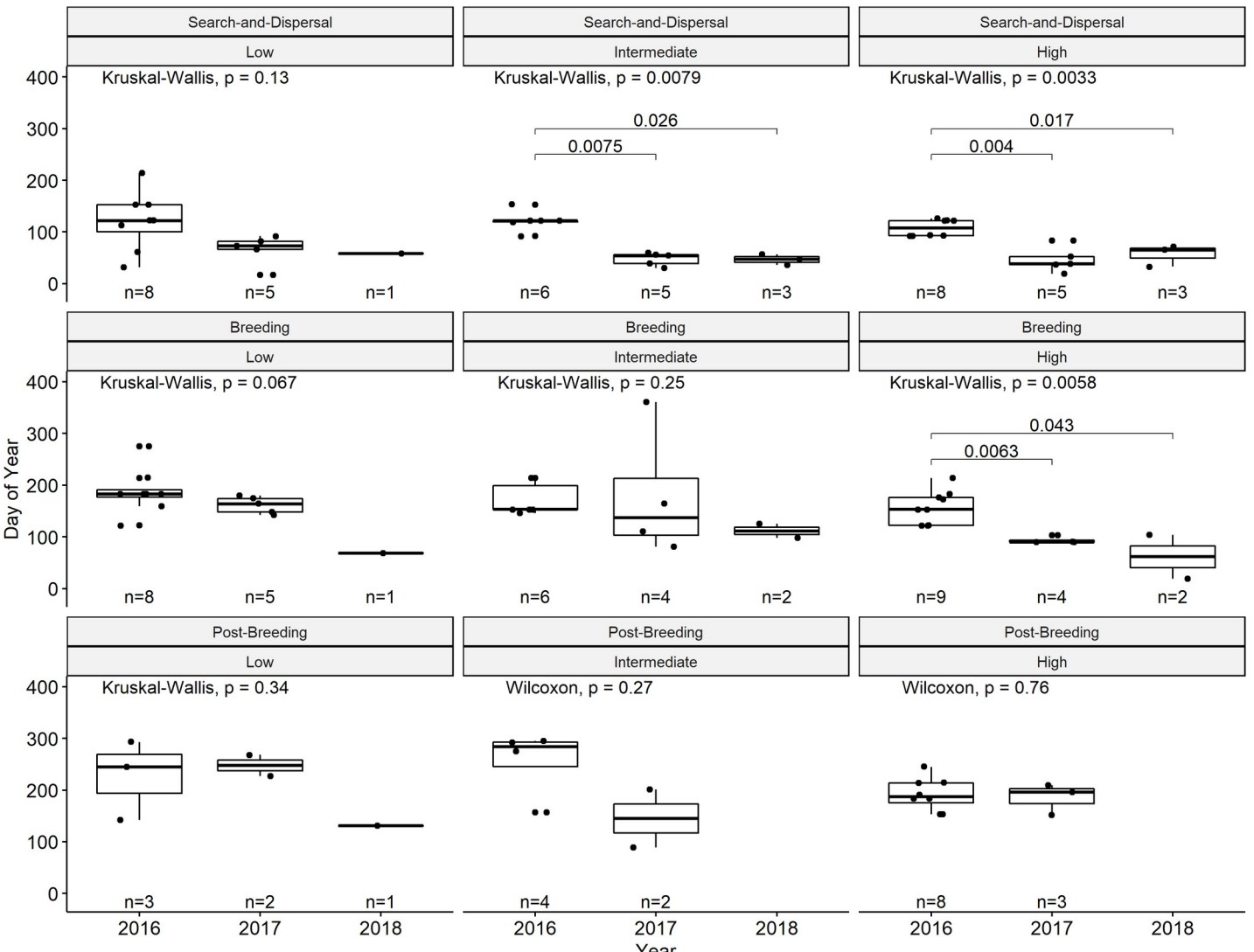

**Fig 5. Boxplots of behavioral season initiation date of search-and-dispersal, breeding, and post-breeding behavioral seasons by year compared within each class (low, intermediate, high) of daytime use of urban habitat during their non-breeding season.** Individual values shown as points in each boxplot. P-values for Kruskal-Wallis tests indicate difference in means within a group, with lines and p-values indicating statistically significant (p < 0.05) Wilcoxon Rank Sum test statistic for respective pairwise differences in means.

Contrary to our prediction, the amount of time spent in non-breeding seasons increased with greater urban habitat use; this concomitantly occurred with earlier initiation of breeding seasons, and direct movement to, or decreased time searching for, breeding areas. Breeding season average duration was 10 weeks; a length of time sufficient to successfully fledge young [18]. Therefore, the observed patterns of longer non-breeding seasons and short or skipped search and dispersal are not necessarily the result of failure to reproduce, but rather are related to urban birds maximizing time spent in their urban ranges by moving directly to known or nearby rookeries.

We found the expression of pre-breeding and breeding seasons shifts with age; older birds spend less time looking for breeding sites and more time exhibiting reproductive behaviors, likely related to increased experience as ibis learn how to better locate rookeries and raise successful clutches as they age. Younger ibis that might not be sexually mature or are unable to

successfully procure mates still exhibit seasonal behaviors throughout the year. Search-and-dispersal seasons of younger ibis, especially in the absence of breeding seasons, are likely evidence of ibis exploring active rookeries or serving as helpers in colonies to gain nesting experience before their own breeding attempts [19].

As birds age, their experience with the environment increases. With migratory birds, studies have demonstrated that older birds are important in modifying established migratory patterns. For example, Teitelbaum *et al.* [46] found that older Whooping cranes (*Grus americana*) were the first to modify post-breeding migratory behavior, instead shortstopping to utilize newly available agricultural overwinter sites with high grain cover. Ibis, similar to whooping cranes, roost and breed in demographically heterogeneous as well as mixed-species flocks, which provide social learning opportunities related to breeding and feeding sites.

Many species exhibit distinct movement patterns for different behavioral states in response to fluctuating resource availability, sociality, mating and breeding requirements, and altered environments [9, 47–49]. Some white ibis in Florida clearly exhibited such movement patterns in response to variability in the timing and location of resources and their innate behavioral plasticity, leading to nomadic movements similar to other species [4, 50, 51]. However, as we clearly observed in our study, animals may respond to altered landscapes through changes in resource use, movements, and even shifts in seasonal behaviors. Previous studies have documented similar changes in movement behaviors in response to partial to full reliance on urban resources [13, 52–54]. For example, Hadeda Ibis (*Bostrychia hagedash*) have colonized urban areas in the Western Cape of South Africa potentially aided by the ameliorating effects on weather and consistent resources provided by urban environments such as planting shade-providing trees and irrigation of fields and lawns [13]. Altered seasonal movement patterns related to anthropogenic resource use as we observed in our study has been documented in other studies [14, 54, 55]. Altered movement behaviors can lead to range restriction [54], reduced migration distance [9, 14], or complete abandonment of migratory movements [13, 14, 55], and altered seasonal timing and length [56, 57].

The long-term, population level consequences of urbanization on nomadic species are not well understood. Altered annual cycles, especially the timing of breeding, movement patterns, and habitat use could contribute to population separation if urban and wildland populations are breeding at different times and in different locations. Urban populations may become isolated from wildland populations through limited knowledge of multiple breeding locations and higher site fidelity to known breeding habitats regardless of quality, and by responding to environmental and social cues that differ from those in wildland habitats [58]. However, the main constraint for urban ibis populations is the availability of suitable rookery sites that, with very few exceptions, only occur in more wildland areas, necessitating movements to breed, and likely continuing to mix urban and wildland populations.

Some researchers have suggested that urban and wildland populations fundamentally differ in their personality and consequently, their behavior, but that the permanence and mechanisms allowing a shift from wild to urban is unclear [58–61]. Yet, studies involving manipulated anthropogenic resource availability show that some populations will revert to wild-type movement patterns when anthropogenic resources such as garbage piles are removed [54], supporting hypotheses of phenotypic plasticity [62]. Social influences could heighten the influence of phenotypic plasticity as more individuals are recruited to the urban population through social learning. Other studies offer evidence indicating contradictory heritable traits between urban and rural populations, suggestive of microevolution processes [58]. If ibis that continue to use wetland areas throughout the year are fundamentally different in their personality (e.g., fear of humans, willingness to explore novel foods), population bifurcation may

occur if wild and urban populations remain separated in breeding seasons, potentially leading to reproductive isolation and permanent adaptation to urban dwelling [63].

For wide ranging species that exhibit distinct movement patterns throughout their annual cycles, GPS data, providing information about fine-scale movement patterns, can be used to infer behavior when direct behavioral observations are not available. From our study, using movement patterns to segment GPS tracks into season behaviors for ibis captured in the same region and experiencing similar environmental characteristics, we determined each ibis was highly variable in the timing and duration of their behavioral seasons. If date or environmental cutoffs had been used to segment data, many sections of the track would have been misrepresented, especially in scenarios when a season was skipped. Behavioral inferences made from GPS tracks are undoubtedly imperfect as we cannot know exact behaviors; however, they are valuable when direct observations cannot be made and the seasonal definitions derived are more accurate than simple calendar cutoffs.

## Conclusions

Animals exhibit individualistic behaviors in response to variable environmental conditions, ongoing adaptation, and potentially reversible behavioral changes, making the study of seasonal behaviors and ecological processes complex. We provide evidence that suggest habitat use, experience, and annual variation can affect the timing and duration of seasonal behaviors. For a nomadic species such as the white ibis, such changes in response to urbanization may eventually lead to phenotypic divergence with resulting social niche separation [49] where urban birds becoming year-round residents and wild birds continuing nomadic movements. The observed differences in the duration and timing of seasons across a range of urban habitat use and ibis age provide evidence that other aspects of ibis ecology such as space use and resource selection may also differ with varying degrees of synanthropic behaviors and across seasons. Similar to studies of many other species, most studies of ibis tend to focus on their ecology while in wildland habitats and on breeding grounds, providing little information about urban, non-breeding, and inter-seasonal ecology [18, 64]. Further studies are needed to understand the influence of anthropogenic land use change on the behavioral changes such as space use, resource selection, potential social carryover effects [65], and the potential for social niche formation [49] by examining full annual cycle ecology of populations existing in landscapes of differing land use composition using animal-derived behavioral season definitions.

## Supporting information

**S1 Fig. Example of a flat summary of BCPA output showing the change in the movement statistic over time for ibis "24_LCS01".** The persistence velocity (y-axis) between consecutive locations is calculated with the BCPA and plotted over time. Vertical lines indicate the significant change points with the width of the lines proportional to the number of times that change point was selected in the moving window analysis. The black and red lines show the mean and standard deviation estimate of the persistence velocity. The coloured circles (ρ hat in the legend) reflect the autocorrelation time scale (Gurarie 2013). Upper panel shows the unfiltered BCPA output depicting every change point selected in the moving window analysis. Lower panel shows the filtered BCPA output that selects significant change points from the neighbouring change points within 10 days.
(TIF)

**S2 Fig. Boxplots of behavioral season duration by ibis age by year.** Distribution of number of days in each behavioral season (non-breeding, search-and-dispersal, breeding attempt, and

post breeding) according to ibis age.
(TIF)

**S3 Fig. Boxplots of behavioral season initiation date by ibis age by year.** Distribution of number of days in each behavioral season (search-and-dispersal, breeding attempt, and post breeding) according to ibis age.
(TIF)

**S4 Fig. Boxplots of behavioral season duration by ibis sex.** Distribution of number of days in each behavioral season (non-breeding, search-and-dispersal, breeding attempt, and post breeding).
(TIF)

**S5 Fig.**
(JPG)

**S1 File. Defining ibis BCPA bouts as behavioral seasons.**
(PDF)

# Acknowledgments

We thank Richard Hall, Betsy Kurimo-Beechuk, Claire Teitelbaum for insightful comments on earlier drafts. Funding was provided by the National Science Foundation EEID grant, (DEB-1518611) and the Georgia Ornithological Society. Additional resources were provided by the Warnell School of Forestry and Natural Resources at the University of Georgia and the USDA National Institute of Food and Agriculture McIntire Stennis program, project accession # 219814.

# Author Contributions

**Conceptualization:** Anjelika Kidd-Weaver, Jeffrey Hepinstall-Cymerman, Catharine N. Welch, Sonia M. Hernandez.

**Data curation:** Anjelika Kidd-Weaver, Jeffrey Hepinstall-Cymerman, Michael J. Yabsley.

**Formal analysis:** Anjelika Kidd-Weaver, Jeffrey Hepinstall-Cymerman, Michael J. Yabsley.

**Funding acquisition:** Sonia M. Hernandez.

**Investigation:** Catharine N. Welch, Maureen H. Murray, Henry C. Adams, Taylor J. Ellison, Sonia M. Hernandez.

**Methodology:** Anjelika Kidd-Weaver, Jeffrey Hepinstall-Cymerman, Catharine N. Welch, Maureen H. Murray, Henry C. Adams, Taylor J. Ellison, Michael J. Yabsley, Sonia M. Hernandez.

**Project administration:** Sonia M. Hernandez.

**Resources:** Anjelika Kidd-Weaver, Jeffrey Hepinstall-Cymerman, Sonia M. Hernandez.

**Supervision:** Jeffrey Hepinstall-Cymerman, Sonia M. Hernandez.

**Writing – original draft:** Anjelika Kidd-Weaver, Jeffrey Hepinstall-Cymerman.

**Writing – review & editing:** Anjelika Kidd-Weaver, Jeffrey Hepinstall-Cymerman, Catharine N. Welch, Maureen H. Murray, Henry C. Adams, Taylor J. Ellison, Michael J. Yabsley, Sonia M. Hernandez.

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
