## [Decision Letter · Decision Letter 0]

28 Oct 2019

PONE-D-19-24467

The movements of a recently urbanized wading bird reveal changes in season timing and length related to resource use

PLOS ONE

Dear Dr. Hepinstall-Cymerman,

Thank you for submitting your manuscript to PLOS ONE. After careful consideration, we feel that it has merit but does not fully meet PLOS ONE’s publication criteria as it currently stands. Therefore, we invite you to submit a revised version of the manuscript that addresses the points raised during the review process.

Academic Editor

The two reviewers raise a number of important issues that need to be addressed before this manuscript can be considered for publication in PlosOne.  I therefore invite you to make major revisions to the manuscript, in the light of the reviewers’ comments.  Please pay close attention to the major issues raised by Reviewer 2.

All points raised by the reviewers must be addressed in the revised manuscript, giving page lines of changes for easy reference, or as rebuttals in an accompanying letter.

We would appreciate receiving your revised manuscript by Dec 12 2019 11:59PM. To enhance the reproducibility of your results, we recommend that if applicable you deposit your laboratory protocols in protocols.io, where a protocol can be assigned its own identifier (DOI) such that it can be cited independently in the future. For instructions see: http://journals.plos.org/plosone/s/submission-guidelines#loc-laboratory-protocols

We look forward to receiving your revised manuscript.

Kind regards,

Maura (Gee) Geraldine Chapman, PhD DSc

Academic Editor

PLOS ONE

Journal Requirements:

1. 

2.  We note that Figure1 in your submission contains map images which may be copyrighted. All PLOS content is published under the Creative Commons Attribution License (CC BY 4.0), which means that the manuscript, images, and Supporting Information files will be freely available online, and any third party is permitted to access, download, copy, distribute, and use these materials in any way, even commercially, with proper attribution. For these reasons, we cannot publish previously copyrighted maps or satellite images created using proprietary data, such as Google software (Google Maps, Street View, and Earth). For more information, see our copyright guidelines: http://journals.plos.org/plosone/s/licenses-and-copyright.

a)    You may seek permission from the original copyright holder of Figure(s) [#] to publish the content specifically under the CC BY 4.0 license.

3. 

In your Data Availability statement, you have not specified where the minimal data set underlying the results described in your manuscript can be found. PLOS defines a study's minimal data set as the underlying data used to reach the conclusions drawn in the manuscript and any additional data required to replicate the reported study findings in their entirety. All PLOS journals require that the minimal data set be made fully available. For more information about our data policy, please see http://journals.plos.org/plosone/s/data-availability.

Additional Editor Comments (if provided):

Academic Editor

The two reviewers raise a number of important issues that need to be addressed before this manuscript can be considered for publication in PlosOne. I therefore invite you to make major revisions to the manuscript, in the light of the reviewers’ comments. Please pay close attention to the major issues raised by Reviewer 2.

All points raised by the reviewers must be addressed in the revised manuscript, giving page lines of changes for easy reference, or as rebuttals in an accompanying letter.

Reviewers' comments:

Reviewer's Responses to Questions

**Comments to the Author**

1. Is the manuscript technically sound, and do the data support the conclusions?

Reviewer #1: Partly

Reviewer #2: No

2. Has the statistical analysis been performed appropriately and rigorously? 

Reviewer #1: Yes

Reviewer #2: I Don't Know

3. Have the authors made all data underlying the findings in their manuscript fully available?

Reviewer #1: Yes

Reviewer #2: Yes

4. Is the manuscript presented in an intelligible fashion and written in standard English?

Reviewer #1: Yes

Reviewer #2: Yes

5. Review Comments to the Author

Reviewer #1: General comments

This is an interesting study with a large sample of GPS tracked individuals. The framing of the study is interesting, assessing seasonal behaviors with respect to underlying habitat use as defined by human modification of the land. I believe the study makes a nice contribution to our understanding of animal behavior in response to human land use modification. However, revisions are required to improve the clarity of the results. Many sentences in the results focus on “urban habitat use” however this phrasing often lacks clarity.

Abstract

Line 23: edit “behavioural” to “behavioral” or vice versa throughout.

Introduction

Line 59 & 62: edit “behavioural” to “behavioral” or vice versa throughout. I’ll not flag all of these, they occur throughout the document.

Line 66-67: suggested edit “…the behavioural state of an individual rather than an arbitrary calendar date…” to “…the behavioral state of individuals, representing the “population”, rather than an arbitrary calendar date…”

Line 66-67: suggested edit “Determining behavioural seasons based on animal movement data is a critical need to advance the understanding movement ecology of animals” to “Determining behavioural seasons, based on animal movement data, is critically needed to advance the understanding the movement ecology of animals” or you could go further: “Determining biological meaningful seasons, based on animal behavior, is needed to advance our understanding of animals movement ecology”.

Line 77-78: What is meant by “urban habitat used” in this sentence: “We predicted that ibis are changing when and where they move related to the amount of urban habitat used.” Do the authors mean “urban habitat modified by humans or does this relate to ibis mobility?” Please clarify.

Methods

Line 110: edit to past tense “In recent decades, ibis living in southern Florida are increasing their use of…” to “In recent decades, ibis living in southern Florida have increased their use of…”.

Line 120: the abbreviation “FL” was not been provided earlier, expand here or add in previous section “Palm Beach County, FL”.

Line 200: edit “separating migratory segments of animal migrations” to “separating segments of animal migrations”.

Results

The first paragraph of the results section is mostly methods, I suggest moving the whole paragraph or most of the detail to the methods.

All Results section: expand the abbreviated seasons SD, NB, BR, PB as they aren’t used that often and spelling the words is much clearer.

Lines 252-255. What does n = 62, n = 54, n = 48 and n = 40 refer to? N = 48 birds were tracked, n = 41 were included in the analyses.

Lines 252-281: This section refers to Figure 2, this figure contains a lot of information which isn’t automatically clear. Clarify that juveniles/sub-adults were “adults” when included with respect to breeding season. I suggest you break this figure into a panel of 6 figures: separate birds by sex (e.g. males on the left females on the right), with individual figures for “low”, “intermediate” and “high” urban ibis use.

Lines 286-289: the first part of this sentence reports a result “Ibis daytime use of urban habitat classes in the NB season ranged from 0.3% to 68.6%...” however, the second part is methods and should be moved “which, to facilitate statistical comparisons, we classified into three quantiles defined by cut points at 28.2% and 54.9% representing “low”, “intermediate”, and “high” urban-use.”

Lines 289-290: the focus of this paper is “seasonal behavior”, this sentence lacks clarity and doesn’t include an assessment of “season”. Given the various timing of deployment, deployment periods, locations and bird age this statement requires further detail to explain what exactly is being tested. “Urban and wetland habitat use were negatively correlated (Pearson Correlation = -0.730).”

Figure’s 3, 4 & 5. The points along the x-axis, within each category (e.g. land use, age), are distributed with respect to an undefined “period”. The x-axis label doesn’t contain enough information for the reader to understand how points within each category vary. Please clarify.

Figure 3. Presumably birds of all ages are included, however juvenile and sub-adults wouldn’t engage in breeding activity, please clarify that these birds have not been included where they were immature.

Figure 3. How are birds “grouped by daytime, non-breeding season urban habitat use classes” when this figure assesses all “Four behavioural seasons (non-breeding, search-and-dispersal, breeding attempt, and post breeding)”?

Lines 308-310: suggested edit to increase clarity (similar edits could be made throughout the results. “Significant differences in season initiation dates by urban habitat use differed only in the BR season (Fig 4) such that ibis with high urban-use began their BR seasons earlier in the year than those with low urban-use.” Revised “Ibis with greater use of urban habitats began their breeding season earlier than ibis with lower use of urban habitats (Fig 4).” Many sentences focus on “urban habitat use” however this phrasing often lacks clarity.

Figure 4. Delete juveniles, their behavior does not relate to “breeding season” and shouldn’t be assessed in this framework, they are immature.

Lines 313-315: The 2018 sample size is questionably low for inclusion in this analysis, I suggest removing it where it is <3 and the authors consider removing 2018 from these analyses all together. “Yearly differences in initiation dates were significantly different for intermediate urban-use ibis only in the SD season, which started significantly later in 2016 than either 2017 or 2018 (Fig 5). For high-urban use ibis, initiation dates differed significantly in both the SD and BR seasons, both starting later in 2016 than 2017 or 2018.”

Lines 332-337: lack clarity, revise to statements.

Lines 340-341: As breeding requires birds to be sexually mature this finding goes without saying. If you are testing adult ibis age then this is meaningful, however the methods do not describe a way to age adult ibis other than to say they are adults. Furthermore, figure 5 does not present age at all. If you are referring to figure 4 then adults are lumped together and compared to juveniles and sub-adults, which does not support this sentence. “Older ibis tended to spend more time dedicated to breeding with mean season duration by age significantly different for SD and BR seasons (Fig 5).”

Lines 341-343: I question the value of this result as “juveniles” were not sexually mature and would not engage in breeding activities. “Juveniles had significantly longer SD seasons compared to both subadults (with 38% subadults (n=8) skipping this season altogether) and adults.”

Lines 343-345: is this biologically meaningful, did sub-adults engage in breeding? “Adults had longer breeding seasons than subadults. Season initiation dates were suggestive of differences in the SD season (p < 0.1) with subadults starting later in the year than adults.” If sub-adults engaged in breeding does that mean they transitioned to be “adults” and thus comparing them with “sub-adults” isn’t possible?

Lines 345-349: this sentence lacks clarity, revise to statements “Seasons skipped also varied with age: three each of the six skipped SD seasons were skipped by subadults (14% of subadult SD seasons skipped) and adults (12%); 3 of skipped breeding seasons were by subadults (38%); 4 of the 14 skipped PB seasons were by subadults (50%), and 10 were skipped by adults 4 (31%).”

Lines 355-358: lack clarity, revise to statements.

Discussion

The Discussion should be revised with respect to the comments on the results e.g. Lines 365-367: consider removing reference to “age” – “We found that season initiation and duration are related to urban habitat use, individual traits such as age and sex, and annual variation likely related to yearly environmental variations.”

Lines 375-379: If this point relates to the time spent by adult vs non-adults then it needs to be revised as this isn’t a valid comparison. “We found the amount of time spent in pre-breeding and breeding seasons shifts with age; older birds spend less time looking for breeding sites and more time exhibiting reproductive behaviours, likely related to increased experience as ibis learn how to better locate rookeries and raise successful clutches as they age.”

Lines 379-382: I’ve been unable to revise the original reference, however I question the stated behavior of “younger ibis serving as helpers”. Have the authors observed this behavior directly associated with “cooperative breeding” as is implied by this sentence? “For younger ibis, longer search-and-dispersal seasons, especially in the absence of breeding seasons, are likely evidence of ibis serving as helpers in colonies to gain nesting experience before their own breeding attempts [19].”

Line 402: as stated above, this sentence may need to be revised – “We found changes in seasonal timing and duration related to age.”

Reviewer #2: I appreciate the time and effort that went into this work but I'm afraid that the manuscript has two major flaws that significantly limit the research. First, the conceptual basis, justification, and objectives of the study are not at all clear. The meaning of "behavioral season" in the Introduction as it relates to movement left me wondering whether all types of movement were being considered or just what one typically thinks of as seasonal movement, i.e. dispersal and migration. This is cleared up in the Methods but needs to be explained much better up front, in the Introduction.

The objectives of the study are to define behavioral seasons and to understand how behavioral seasons vary with respect to habitat type (wild vs urban). The justification for the first objective eludes me. How does identifying individual season start and end dates help us? Of course, season duration and timing will vary among individuals. This is a basic tenet of niche ecology and the data presented in this manuscript exemplify it well. On the other hand, using environmental cues to define seasons is useful. Environmental cues vary regionally and researchers are thereby not misled by date. This is the approach used by large-scale citizen science programs and works well. If the purpose of the study is to avoid the problems described at lines 444-446, then this is useful but the study then becomes a methodological one and the Introduction should be framed accordingly.

The justification for the second objective is just as unclear. What do we know about urbanization and the timing and duration of species phenology as it relates to birds? Quite a bit, including what is described at lines 112-114. What does this study add to existing knowledge? Earlier onset of breeding has been demonstrated for many species in urban areas. The shorter dispersal period of ibis in urban habitats in the study area is interesting and potentially novel, but again, the authors should re-frame the paper to make this more of a focus if that is the intent.

Second, I'm not convinced that the study design is adequate to address the objectives. It appears that different birds were tracked during different seasons and years. It would be ideal to track multiple individuals throughout all seasons in all years (or over one year) to avoid the possibility that results in one season, ie season duration, are driven by a few idiosyncratic individuals. It's not at all clear whether individual is the correct unit of analysis to analyze the effect of habitat type on season duration and timing. Do individuals within a site influence each others' movement? Finally, how does inter-annual variation affect the results? Are habitat type and year confounded? Some of these concerns may be unfounded but I found it very difficult to determine if this was the case.

Two more minor comments: 1. The description of the definition of behavioral seasons at lines 216-224 seems very objective. I have a hard time seeing someone else being able to replicate these methods. Does this significantly impair the utility of the behavioral season concept?; 2. The Discussion section is mostly inference that is very weakly based on the results. Too much is being inferred from the results as they stand.

6. PLOS authors have the option to publish the peer review history of their article (what does this mean?). If published, this will include your full peer review and any attached files.

Reviewer #1: No

Reviewer #2: No

---

## [Author Response · Author response to Decision Letter 0]

29 Dec 2019

We have uploaded a document containing all of the Response to Editor and Reviewer comments.

---

## [Decision Letter · Decision Letter 1]

24 Feb 2020

The movements of a recently urbanized wading bird reveal changes in season timing and length related to resource use

PONE-D-19-24467R1

Dear Dr. Hepinstall-Cymerman,

We are pleased to inform you that your manuscript has been judged scientifically suitable for publication and will be formally accepted for publication once it complies with all outstanding technical requirements.

With kind regards,

Maura (Gee) Geraldine Chapman, PhD DSc

Academic Editor

PLOS ONE

Additional Editor Comments (optional):

Reviewers' comments:

Reviewer's Responses to Questions

**Comments to the Author**

1. If the authors have adequately addressed your comments raised in a previous round of review and you feel that this manuscript is now acceptable for publication, you may indicate that here to bypass the “Comments to the Author” section, enter your conflict of interest statement in the “Confidential to Editor” section, and submit your "Accept" recommendation.

Reviewer #2: All comments have been addressed

2. Is the manuscript technically sound, and do the data support the conclusions?

Reviewer #2: (No Response)

3. Has the statistical analysis been performed appropriately and rigorously? 

Reviewer #2: (No Response)

4. Have the authors made all data underlying the findings in their manuscript fully available?

Reviewer #2: (No Response)

5. Is the manuscript presented in an intelligible fashion and written in standard English?

Reviewer #2: (No Response)

6. Review Comments to the Author

Reviewer #2: (No Response)

7. PLOS authors have the option to publish the peer review history of their article (what does this mean?). If published, this will include your full peer review and any attached files.

Reviewer #2: No

---

## [Editor Report · Acceptance letter]

9 Mar 2020

PONE-D-19-24467R1 

The movements of a recently urbanized wading bird reveal changes in season timing and length related to resource use 

Dear Dr. Hepinstall-Cymerman:

I am pleased to inform you that your manuscript has been deemed suitable for publication in PLOS ONE. Congratulations! Your manuscript is now with our production department. 

With kind regards,

on behalf of

Professor Maura (Gee) Geraldine Chapman 

Academic Editor

PLOS ONE